# Role of Ultrasonography in the Diagnosis of Wilms’ Tumour

**DOI:** 10.3390/children9081252

**Published:** 2022-08-19

**Authors:** Radu Ninel Bălănescu, Alexandru Emil Băetu, Andreea Alecsandra Moga, Laura Bălănescu

**Affiliations:** 1“Grigore Alexandrescu” Clinical Emergency Hospital for Children, 011743 Bucharest, Romania; 2Department of Pediatric Surgery and Anaesthesia and Intensive Care, “Carol Davila” University of Medicine and Pharmacy, 020021 Bucharest, Romania

**Keywords:** Wilms’ tumour, ultrasound, Doppler, computed tomography

## Abstract

Background. Wilms’ tumour or nephroblastoma is the most common renal malignancy encountered in the paediatric population. Imaging findings are of great importance to the surgeon, the oncologist and the radiologist in the diagnosis and the staging and surveillance of this tumour. Material and Methods. This study was carried out as a 10-year retrospective study of patients who were diagnosed with Wilms’ tumour. Results. The study included 12 boys and 11 girls. Ultrasound and computed tomography were performed in all cases. Ultrasonography was found to be superior to the CT examination when approximating the antero-posterior and transverse diameters; the computer-tomographic examination is cited as superior for estimating the invasion of nephroblastomas. Conclusions: Ultrasound has been shown to be effective in detecting the rupture of the renal capsule, tumour calcifications and invasion of the renal vein, pelvis and ureter. However, ultrasound cannot replace CT in the detection of lymphadenopathy and the invasion of adjacent organs.

## 1. Introduction

Wilms’ tumour (WT) or nephroblastoma is the most common renal malignancy in the paediatric population, with about 1000 new cases being diagnosed in Europe each year. In total, 95% of all renal cancers and 6% of cancers in children under the age of 15 are found to be nephroblastomas [1,2,3].

WT tends to grow into the vena cava and can sometimes even invade the atrium, which usually increases intraoperative complication rates [4]. The 5-year overall survival for a patient with WT is excellent (90%), due to optimised use of current treatment strategies, including nephrectomy, chemotherapy and, in certain cases, even radiotherapy [2,5,6,7].

Based on previous experience, the UMBRELLA SIOP-RTSG 2016 calls for the management of patients with renal neoplasm with initial chemotherapy without biopsy followed by surgery and adjuvant therapy with chemotherapy and/or radiation. This being the case, imaging studies play an important role in determining the stage of the neoplasm and establishing the diagnosis of WT, minimising the risk of using chemotherapy in treating patients with lesions different from WT [6,7,8].

Imaging findings are of great importance to the surgeon, the oncologist and the radiologist in the diagnosis and to the staging and surveillance of WT. Initial imaging of a renal mass begins with an abdominal ultrasound (US) that identifies the point of origin, which can sometimes be challenging due to the size of the tumour mass [8,9].

The advantage of using ultrasound in the initial diagnosis of WT is that this a non-invasive imaging method that can also be applicable at the patient’s bedside whenever necessary and does not require patient sedation. Doppler US is also useful when examining the patency of the renal vein and the inferior vena cava (IVC) while searching for tumour thrombus [6,7,10].

According to the UMBRELLA protocol, magnetic resonance imaging (MRI) of the abdomen is the first choice in complementary imaging, as this technique is not associated with ionizing hazards as is the case when using computed tomography (CT) [6]. However, the decision in choosing either a CT scan or an MRI for abdominal staging of a WT is at the discretion of each institution, both having similar diagnostic performance in the staging of WT [8].

We retrospectively reviewed the medical records of 23 patients diagnosed and surgically treated for nephroblastoma at our institution in order to assess the value of the two imaging modalities, ultrasound and computed tomography, in the initial evaluation of nephroblastomas. The clinical translation of the paper consists of the evaluation of the possibility of using a non-invasive, non-irradiating method as the initial imagistic tool for the diagnosis and staging of nephroblastoma or whether it is of greater sensibility in the evaluation of the essential imagistic characteristics of nephroblastomas, such as tumour dimension, bilaterality, vascular invasion, pelvic or ureteral extension, renal capsule involvement, invasion of adjacent organs, lymph node involvement and the presence of metastases. We hypothesise that ultrasonography can be of great value in the diagnosis and evaluation of tumour dimensions and local extension, but computed tomography is mandatory for the accurate staging of nephroblastoma in children.

## 2. Aim

The aim of this study is to elucidate the place and the advantages that ultrasonography has for patients with Wilms’ tumour compared to computed tomography, using as state-of-the-art intraoperative examination and histopathological examination.

## 3. Materials and Methods

The present study is an analytical, observational study involving a cohort of 23 patients with Wilms’ tumours whose data were collected retrospectively over a period of 10 years from the medical archive of the “Grigore Alexandrescu” Emergency Hospital. Despite the fact that the small number of patients is an obvious shortcoming, this cohort belonged to a single surgical team, which greatly reduces the variability and heterogeneity often present in surgical studies. Additionally, it is worth mentioning that in the literature most studies involving Wilms’ tumour include a small number of patients.

The inclusion criteria in this study were the presence of Wilms’ tumour in children who were investigated by laboratory analysis and of course imaging by ultrasonography and computed tomography. Children who were insufficiently investigated or those who were lost to follow-up were excluded from the study.

In most cases of clinical examination, an intraabdominal tumour mass was identified. Ultrasound was then performed in order to confirm initial diagnosis and to establish the origin of the tumour mass. Contralateral kidney was also evaluated and Doppler US was performed in order to assess for vascular invasion. Patients then underwent chest/abdominal/pelvic contrast-enhanced CT scan and, depending on the age, in certain cases required sedation.

The statistical analysis was performed with GraphPad 9. For each data set analysed, the normality of the distribution was tested with the Anderson–Darling, D’Agostino–Pearson, Shapiro–Wilk and Kolmogorov–Smirnov tests. We used the one sample *t* test to determine the mean with standard deviation, standard error and median. To determine the difference between three or more averages, we used the ANOVA test for repeated measurements with Bonferroni correction. The correlation analysis took into account the data distribution, thus choosing the Pearson or Spearman coefficient and the Bland–Altman test was used to compare imaging methods. The correlation coefficients were represented by a heat-map. To calculate the differences between the distribution frequencies we used Chi-Square test, and the graphical representations were exported from the statistical program.

## 4. Results

We reviewed the records of 23 patients who were diagnosed and underwent surgical treatment for nephroblastoma at our centre. The median age at diagnosis was 3 years ranging from 0 to 11 years. There were 16 patients (70%) from rural areas and 6 (30%) from urban areas. There were 15 patients that presented with abdominal pain, 14 with palpable abdominal mass, 9 with weight loss and 6 with haematuria. Less frequent manifestations were fever, loss of appetite, vomiting and acute urine retention. Thoraco-abdominopelvic computed tomography (CT) studies were used for diagnosis and staging in all cases. Abdominal ultrasound was also used in all cases as the initial diagnostic method. For CT examination, various elements were analysed for precise tumour staging: tumour dimension, bilaterality, vascular invasion, pelvic or ureteral extension, renal capsule involvement, invasion of adjacent organs, lymph node involvement and the presence of metastases. All patients underwent surgery at our centre. Nephrectomy was performed in 21 patients (91%), and partial nephrectomy in 2 patients (9%). In most cases the entire tumour was received as a resection specimen and histologically studied. Three tumours had a biphasic pattern and the majority, 14, had a triphasic pattern. In all monophasic tumours, blastemal cells were predominant. Based on predominant cell types, tumours were classified into low, intermediate or high risk. Most patients had an intermediate risk 14 (60%), while low risk was attributed to only 3 patients (13%) and high risk to 6 patients (26%). Most of the tumours were classified as stage III 10 (43.47%), followed by stage II and stage IV, 21.73% each. Two children had stage I nephroblastoma (8.69%). Only one patient had bilateral nephroblastoma—stage V (4.34%). All patients received adjuvant chemotherapy and seven received adjuvant radiotherapy.

The present study included 12 boys and 11 girls, whose age values expressed in months have a normal distribution in all tests applied. The average age of the boys included in the study was 45.42 months vs. 46.91 months, the difference between the averages (1.4 months) represented with green in the graphic representation, being obviously statistically insignificant *p* = 0.91 (Figure 1). Based on our findings on age and gender distribution, we can suggest that a retroperitoneal tumour in a child of 45.42–46.91 months is more likely to be nephroblastoma and requires looking for its imagistic characteristics, but with no difference between genders.

In order to respond to the main objective of this study, it was necessary to compare the antero-posterior and transverse diameter of the tumours obtained using computed tomography and ultrasound. The measurements considered to be of reference were those performed intraoperatively. For the following analysis we used the ANOVA analysis for repeated measurements and the Bonferroni correction [Table 1].

From the analysis of the differences between the mean value of the antero-posterior diameter, we can say that both CT (85.08 mm) and ultrasonography (92 mm) tend to underestimate the actual diameter of the tumour measured intraoperatively (118.47 mm). This difference is statistically significant *p* = 0.046 [Figure 2].

The one-way ANOVA analysis performed for the transverse diameter of the tumour reveals an interesting result when comparing the three mean values obtained by computed tomography, ultrasonography and intraoperative. The transverse diameter seems to be overestimated using CT and underestimated using ultrasonography (94.27 mm and 67.95 mm) if we compare it with the intraoperative measurements (81.81 mm). The analysis is not statistically significant *p* = 0.054 [Figure 3].

However, in order to see for sure whether ultrasonography can give reference results for nephroblastomas, we made two correlation matrices, represented graphically by a heat-map. For this analysis, the data distribution was taken into account, so that, as appropriate, the Pearson or Spearman correlation coefficient was chosen (Figure 4).

Regarding the transverse diameter, the dimensions found using ultrasonography correlate better than those revealed by the CT examination (*r* = 0.73 *p* = 0.003 vs. *r* = 0.46 *p* = 0.038). The superior correlations of the dimensions from ultrasonography with those measured intraoperatively remain superior in the case of the antero-posterior diameter (*r* = 0.59, *p* = 0.019 vs. *r* = 0.43, *p* = 0.044). In our study, ultrasonography is superior to CT examination when approximating the antero-posterior and transverse diameters, and the computer-tomographic examination is cited as superior for estimating the invasiveness of nephroblastomas.

We performed a Bland–Altman analysis to compare the two methods of estimating the antero-posterior diameter of nephroblastomas (ultrasound and computed tomography) using the intraoperatively measured diameter as a reference value. To calculate the measurement bias, two analyses were performed with the same calculation procedure (ultrasound dimension minus intraoperatively dimension related to the average of these values). Both ultrasonography (bias = −26.26 mm, SD = 69.18 mm with a 95% CI of agreement between −161 mm and 109.3 mm) and computed tomography (bias = −28.35 mm, SD = 79.18 mm with 95% CI of agreement between −183.5 mm and 126.8 mm) tend to systematically underestimate the real value of the anterior-posterior diameter. (Figure 5 and Figure 6).

The Bland–Altman analysis comparing the transversal diameter measurements obtained using ultrasound and CT with that obtained intraoperatively reveals that ultrasonography tends to underestimate while computed tomography tends to systematically overestimate. The bias of ultrasonography is −17.65 mm (SD = 39.36 mm and 95% CI of agreement is between −94.79 and 59.49 mm). The computed tomography bias is 20.04 mm (SD = 51.49 mm and 95% CI of agreement between −80.88 mm and 121 mm) (Figure 7 and Figure 8).

In order to increase the accuracy of the statistical analysis regarding the invasion of nephroblastomas, this time we used as a reference the histopathological examination of the tumour, which includes three essential characteristics: rupture of the renal capsule, invasion of the pelvis and ureter and invasion of the renal vein.

Ultrasonography failed to detect any case of rupture of the renal capsule among the 10 objectified in the histopathological examination. Instead, the computer-tomographic examination still highlighted five of these cases (Figure 9, Table 2). Additionally, ultrasonography failed to detect any case of invasion of the pelvis and ureter among the six existing in our cohort and computed tomography highlighted five of them with great accuracy (Figure 10. Regarding the vascular invasion in the renal vein, 4 of the 11 histopathological cases were detected by ultrasonography and computed tomography revealed 5 cases. However, the distribution frequency of the detection of this vascular invasion is statistically significantly different (*p* = 0.02) from the histopathological one, which makes ultrasonography a poor diagnostic method (Figure 11).

## 5. Discussion

According to the SIOP guidelines, the diagnosis of WT is based on clinical and radiological features, histological examination not being required in order to initiate treatment [3].

Imaging of a renal mass includes an abdominal ultrasound in order to identify the point of origin, followed by contrast-enhanced chest/abdominal/pelvis CT scan or MRI in order to further evaluate the primary site, to identify potential metastases and to establish a preoperative plan [9,11]. In this study, we aimed to establish the additional advantages of ultrasonography in the management of nephroblastoma cases, besides establishing the organ of origin.

Patients with several congenital anomalies such as WAGR syndrome (Wilms’ tumour, aniridia, genitourinary abnormalities, intellectual disability, WT1 gene), Denys-Drash syndrome, Beckwith-Wiedermann syndrome or isolated hemihypertrophy are at higher risk of developing WT [3]. In our study, associated anomalies were found in only one patient who had Beckwith-Wiedemann syndrome. This being the case, abdominal ultrasound is the study of choice in the screening of such patients, surveillance in these cases being performed from the time of diagnosis till the age of 5 every 3–4 months. While Edmund et al. concluded that screening patients with a high risk of WT did not significantly improve survival, unscreened patients are said to present at a later stage [12]. Smaller tumours found due this “aggressive” screening strategy are also more amenable to renal sparing surgery [13,14,15].

When trying to identify if the origin of the tumour mass is the kidney, radiologists often observe the movement of the mass with the kidney during normal breathing. However, one should be aware that this is not the case when the tumour mass infiltrates adjacent organs. US can determine initial staging by detecting possible lymph nodes metastases, hepatic or intraperitoneal lesions and the presence or absence of abdominal or pelvic fluid when examining the peritoneal recesses. US can distinguish between a solid and a cystic mass. It also allows for contralateral kidney evaluation in cases of bilateral tumours and can detect congenital renal or genitourinary malformations that may affect renal function [6,8]. We performed abdominal US in all cases as the initial imagistic method.

On an ultrasound, WT appears as a large, solid intrarenal mass with smooth and well-defined margins, with a pseudo capsule and areas of heterogeneous echogenicity (tumour necrosis) and hypoechoic and anechoic cystic areas. Renal masses can distort the normal parenchyma, with a “claw sign” of the normal parenchyma surrounding the mass and calyceal dilatations may also be present due to the compressive effect of the tumour mass [6,8].

It is of absolute importance that once the diagnosis of a renal mass is established the radiologist evaluates the renal vein and the inferior vena cava with both grey-scale and colour Doppler due to the vascular extension of the tumour [8]. Ultrasound has become the first choice when establishing the diagnosis of WT, with a reported accuracy of 60–100% in detecting tumour thrombi [14]. Ultrasound also allows the evaluation of the degree of adhesions between the thrombus and the vascular wall [4]. In 6–10% of cases, vascular extension with tumour thrombi in the IVC can found when the diagnosis of WT is established [4,9]. Regarding the vascular invasion in the renal vein, 4 out of the 11 histopathological cases were detected by ultrasonography, while computed tomography revealed 5 cases. In their study, Khanna et al. report a lower sensitivity for the detection of thrombi on ultrasound of 70% compared to 84.6% on CT scan [16,17]. However, intraoperative IVC and/or renal vein palpation is essential in order to avoid transecting an unidentified thrombus [9]. We found that the distribution frequency of the detection of this vascular invasion is statistically significantly different (*p* = 0.02) from the histopathological one, which makes ultrasonography a poor diagnostic method. Due to the risk of tumour rupture, tumour outline, the presence of hyperechoic areas (that may represent haemorrhage) and the presence of blood in the abdominal cavity should be carefully assessed [6].

In some cases, WT can completely obstruct the IVC without invasion. US can better and more accurately assess the IVC in multiple planes and degrees of obliquity, as the radiologist is able to appreciate if the tumour is simply compressing the IVC or actually invading it [18].

An important drawback of US that was shown in previous studies was the less than accurate tumour staging when compared to a CT scan [19]. In our study, imagistic staging was carried out by computed tomography examination, and we found that most of the tumours were classified as stage III 10 (43.47%), followed by stage II and stage IV, 21.73% each. Two children had stage I nephroblastomas (8.69%). Only one patient had bilateral nephroblastoma—stage V (4.34%). However, recent studies have shown that US plays an important role in the differential diagnosis, in staging, monitoring and surgical planning when dealing with renal tumours [18].

After the renal mass is confirmed on an ultrasound, cross-sectional chest/abdominal/pelvis CT or MRI is then further used in order to evaluate the primary site, the status of the contralateral kidney, tumour involvement of renal veins or inferior vena cava, to identify possible metastases, preoperative tumour rupture and the existence of ascites. CT evaluation of the lungs is mandatory in order to assess for small lung nodules that may not be visible on radiograph, known as CT-only nodules. Administration of intravenous contrast is also mandatory in order to fully evaluate the renal hilum and the potential intravascular thrombi and usually only one portal venous phase is sufficient for the diagnosis of WT. Delayed-phase imaging can help when evaluating the status of the renal collecting system with regard to the tumour, especially when dealing with a small tumour, in which case nephron-sparing surgery can be performed [8,9].

On a CT scan, WT appears as a heterogenous soft-tissue mass that shows less enhancement than normal renal parenchyma. Enhancement is also patchy and allows for a better delineation between the tumour and the kidney. WT usually presents with hypodense areas due to necrosis, old haemorrhage and cysts, and in some cases, calcifications, infiltration and the distortion of the calyceal system, vascular invasion and lymph node metastases can also be found [6]. Some tumours may grow in an exophytic manner with the bulk of the tumour being seen outside the renal cortex [19].

In their study, Le Rouzic et al. reported that a CT scan has a good sensitivity in detecting acute haemorrhage. Tumour necrosis, perinephric extension, lymph node involvement were also more often detected on a CT scan then on US [18]. In our cohort, we also found that both ultrasonography and computed tomography tend to systematically underestimate the real value of the anterior-posterior diameter, while transversal diameter measurements obtained by ultrasound and CT with that obtained intraoperatively reveals that ultrasonography tends to underestimate while computed tomography tends to systematically overestimate. Additionally, ultrasonography failed to detect any case of rupture of the renal capsule among the 10 objectified in the histopathological examination. Instead, the computer-tomographic examination still highlighted five of these cases.

It is certain that patients with WT require life-long follow-up as most of them end up having a single kidney and there is still a risk of developing late effects of treatment or secondary tumours (the risk of a second neoplasm is 5–7% at the 30-year mark-up) [3].

According to the SIOP guidelines, ultrasound examination should be performed 5–8 times during the first 5 years after treatment in single kidney disease with no metastases, in order to detect possible tumour recurrences [20]. About 15% of patients treated for WT relapse, more often at the 2-year postoperative mark and only occasionally after 5 years (0.5% of cases), with the most common sites of relapse being the lung, the abdomen and the liver. Bone and brain involvement is rare [2,10]. Survival after relapse is stratified based on risk factors and is reported as 34–64% [2,3].

The best method of treatement and the surveillance for patients with WT after treatment are still under debate. Strategies vary in different centres, depending on the available resources and can sometimes be a burden both for the family and for the healthcare system [2].

Bearing that in mind, ultrasonographic examination is of great value in the prompt detection of abdominal relapse as it is a non-invasive imaging method that can also be applicable whenever necessary, and when compared with a CT scan or an MRI, is a more inexpensive investigation method [10]. Nonetheless, equivocal findings using an ultrasound require additional CT or MRI imaging. Earlier studies have shown that a CT scan is not necessary in most cases of WT relapse, clinical, ultrasonographic findings being sufficient in establishing a correct diagnosis.

## 6. Conclusions

In our clinic, ultrasonography tends to be superior to computed tomographic examination in terms of measuring the antero-posterior and transverse diameter, but it is inferior in the assessment of neoplastic invasion (rupture of renal capsule, invasion of pelvis, ureter and renal vein).

## 7. Study Limitations

Our study has several limitations. First, it is a retrospective study from a single centre, with a small number of cases. Preoperative imagistic evaluation of the patients was performed by different members of the radiological team of our institution, resulting in variations in describing the key elements of the preoperative assessment of nephroblastoma.

## Figures and Tables

**Figure 1 children-09-01252-f001:**
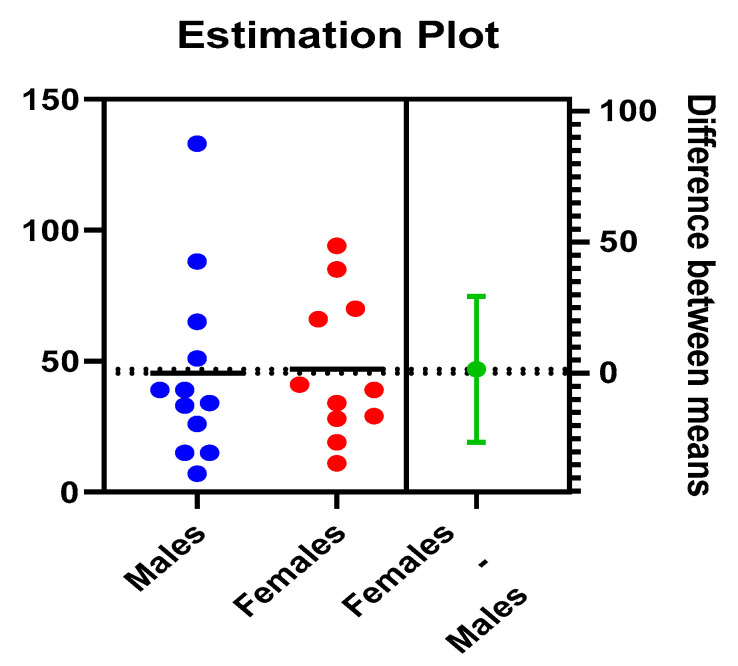
Gender and age distribution in nephroblastoma.

**Figure 2 children-09-01252-f002:**
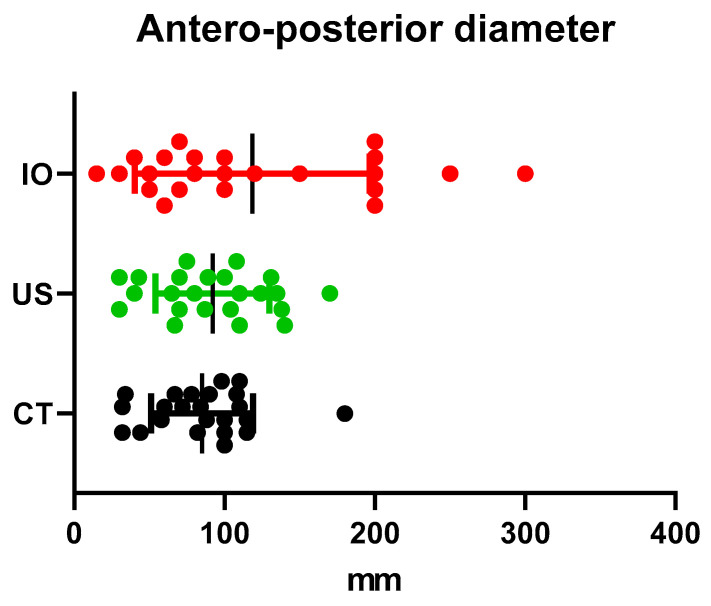
Antero-posterior diameter comparison using US and CT (US = ultrasound, CT = computed tomography, IO = intraoperatively).

**Figure 3 children-09-01252-f003:**
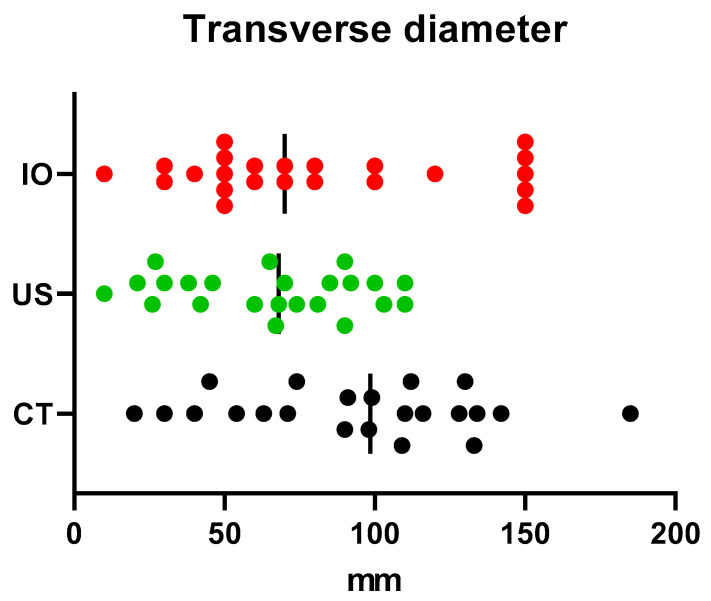
Transverse diameter comparison using US and CT (US = ultrasound, CT = computed tomography, IO = intraoperatively).

**Figure 4 children-09-01252-f004:**
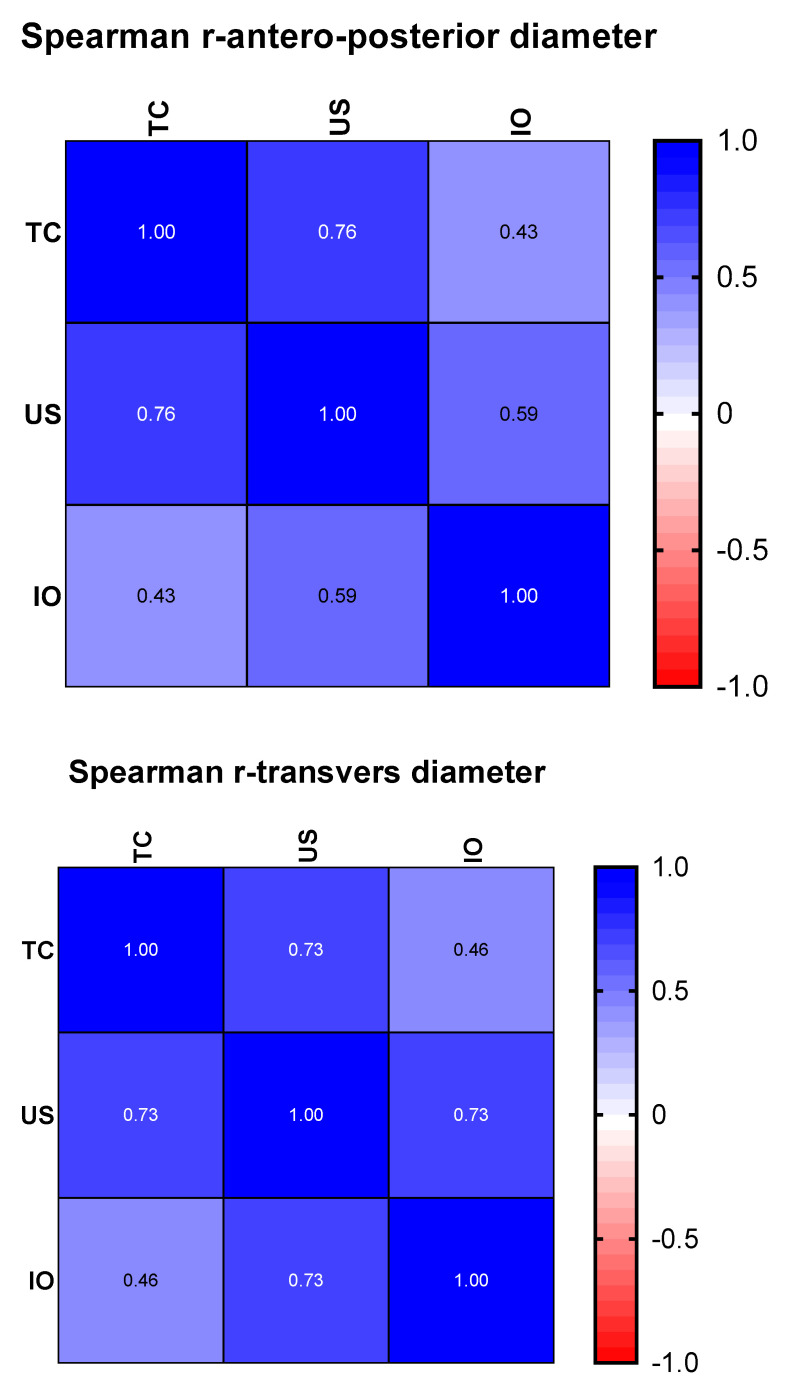
Spearman correlation coefficient (US = ultrasound, TC = computed tomography, IO = intraoperatively).

**Figure 5 children-09-01252-f005:**
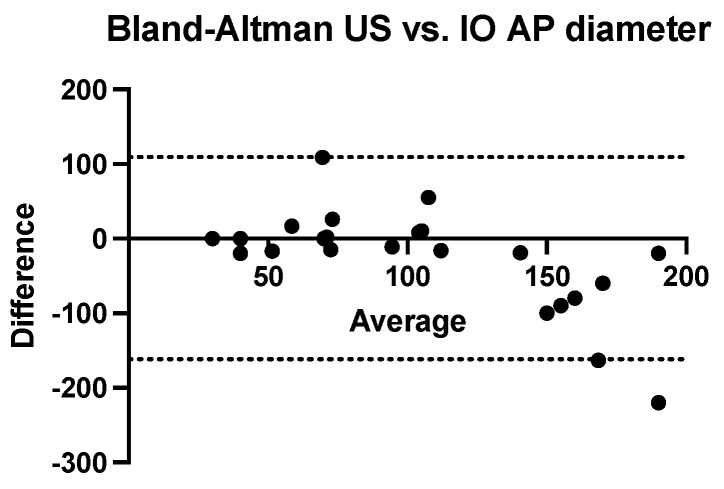
Bland-Altman analysis comparing anteroposterior diameter measured using ultrasound versus intraoperative measurement (US = ultrasound, IO = intraoperatively).

**Figure 6 children-09-01252-f006:**
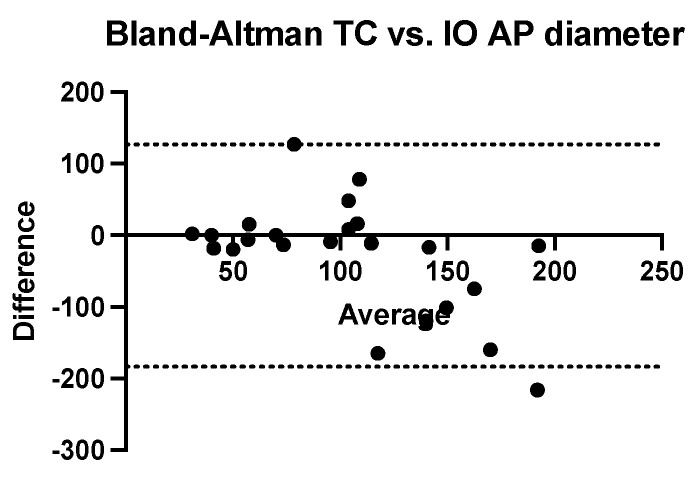
Bland-Altman analysis comparing anteroposterior diameter measured using computed tomography versus intraoperative measurement (TC = computed tomography, IO = intraoperatively).

**Figure 7 children-09-01252-f007:**
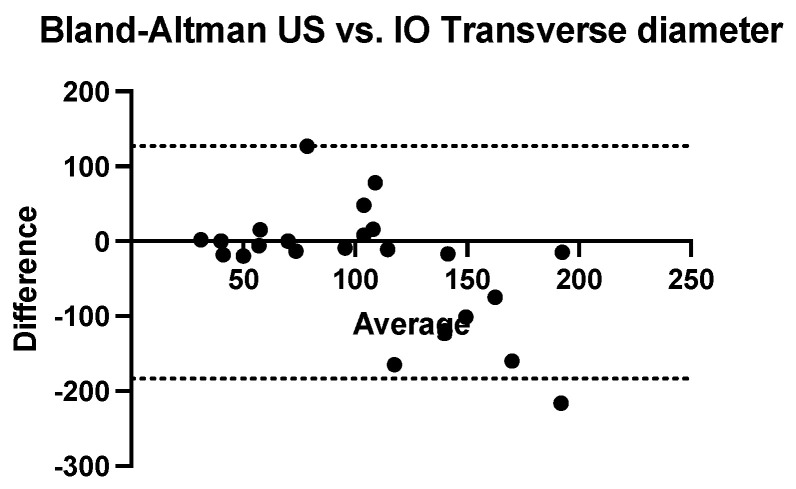
Bland-Altman analysis comparing transverse diameter measured using ultrasound versus intraoperative measurement (US = ultrasound, IO = intraoperatively).

**Figure 8 children-09-01252-f008:**
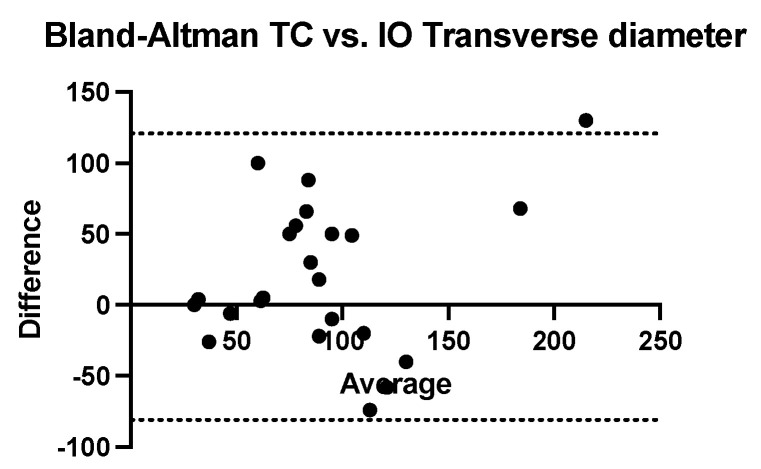
Bland-Altman analysis comparing transverse diameter measured using com-puted tomography versus intraoperative measurement (TC = computed tomography, IO = in-traoperatively).

**Figure 9 children-09-01252-f009:**
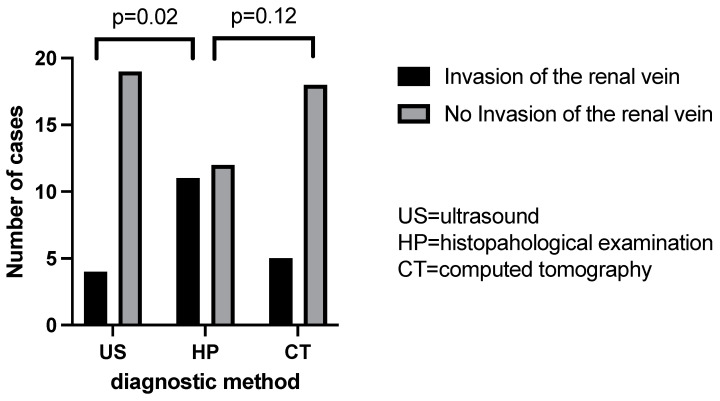
Renal vein invasion evaluation by US/CT comparative to histopathological examination.

**Figure 10 children-09-01252-f010:**
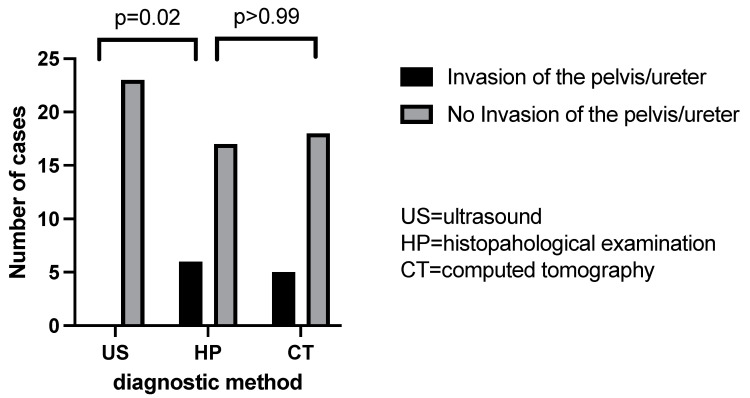
Pelvis/ureteral invasion evaluation by US/CT comparative to histopathological examination.

**Figure 11 children-09-01252-f011:**
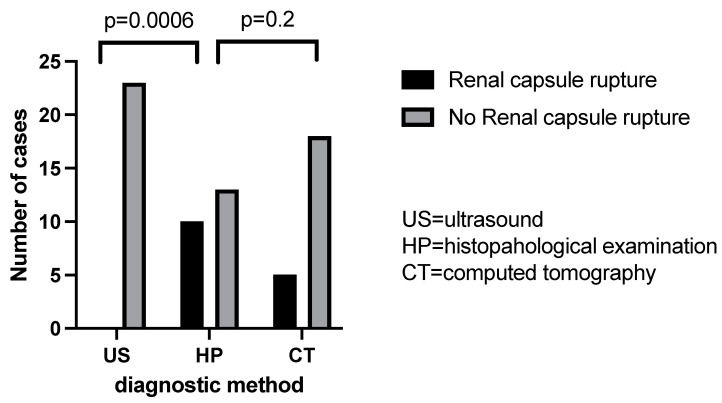
Renal capsule invasion evaluation by US/CT comparative to histopathological examination.

**Table 1 children-09-01252-t001:** Comparison between antero-posterior and transvers diameter using US and CT.

Parameter	Mean Value CT (mm)	95% CI-CT	Mean Value US (mm)	95% CI-US	Mean Value Intraoperatively (mm)	95% CI-IO	*p* Value
Antero-posterior diameter	850,870	704,260 to 997,479	920,000	756,132 to 1,083,868	1,184,783	847,204 to 1,522,362	0.046
Transverse diameter	942,727	759,432 to 1,126,022	679,545	553,207 to 805,884	818,182	617,722 to 1,018,642	0.054

**Table 2 children-09-01252-t002:** Chi square analysis (US = ultrasound, TC = computed tomography, IO = intraoperatively).

	US Percentage and Number of Detection	TC Percentage and Number of Detection	HP Percentage and Number of Detection	US vs. HP (*p* Value)	TC vs. HP (*p* Value)
Renal capsule rupture	0%	21.74 (n = 5)	43.48% (n = 10)	0.0006	0.2
Invasion of the pelvis/ureter	0%	21.74% (n = 5)	26.09% (n = 6)	0.02	0.99
Invasion of the renal vein	17.39% (n = 4)	21.74% (n = 5)	47.83% (n = 11)	0.02	0.12

## Data Availability

Data is available on the following email address: alexandru.baetu@gmail.com.

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
