# Peer review of "Role of Ultrasonography in the Diagnosis of Wilms’ Tumour"

_children, 2022, doi:10.3390/children9081252_

Round 1
Reviewer 1 Report
The paper describes a the experience with Ultrasound (US) and Computed Tomography (CT) in a 10-year retrospective cohort of patients suspected of Wilms tumors. In particular the authors identify the areas where US or CT is preferable in detecting treatment policy changing features.
The authors present few details on the patients (age and sex) and a detailed list of laboratory values. It is not entirely clear why these lab values with their distribution characteristics are presented. The data are not further commented. It is also not clear why the age between boys and girls is tested. No hypothesis in respect is mentioned. However, it would be nice to get some more detailed information on the histology of the tumors as well as potential congenital anomalies.
The analysis on the agreement between two different methods (and the reference) are not presented optimally. Figure 2 and 3 and the correlation plot do not capture the relatedness of the measurements. The distribution on itself are only part of the story as well as the correlation plots. However, a high correlation does not necessarily imply that there is good agreement between the two methods. The Bland–Altman plot (difference plot) is a method of data plotting that covers better the agreement between two different methods. It can also be used to compare a new measurement technique or method with a gold standard, as even a gold standard does not—and should not—imply it to be without error.
In this respect, also the inter-rater agreement is difficult to follow. It is unclear what measurement of estimate the investigators used for their analysis. By only giving test results it is impossible to appreciate what the investigators have done or whether it makes sense.
Author Response
Reviewer 1
The paper describes the experience with Ultrasound (US) and Computed Tomography (CT) in a 10-year retrospective cohort of patients suspected of Wilms tumors. In particular the authors identify the areas where US or CT is preferable in detecting treatment policy changing features.
Point 1: The authors present few details on the patients (age and sex) and a detailed list of laboratory values. It is not entirely clear why these lab values with their distribution characteristics are presented. The data are not further commented. It is also not clear why the age between boys and girls is tested. No hypothesis in respect is mentioned. However, it would be nice to get some more detailed information on the histology of the tumors as well as potential congenital anomalies.
Response 1: Thank you for your suggestion, we will add a paragraph about the demographic data of the patients included in our study. Also, clinical and histological characteristics of these cases will be described. Based on our findings on age and gender distribution, we can suggest that a retroperitoneal tumor in a children of 45.42/ 46.91 months is more likely to be nephroblastoma and look for its imagistic characteristics, but with no difference between genders.
Point 2: The analysis on the agreement between two different methods (and the reference) are not presented optimally. Figure 2 and 3 and the correlation plot do not capture the relatedness of the measurements. The distribution on itself are only part of the story as well as the correlation plots. However, a high correlation does not necessarily imply that there is good agreement between the two methods. The Bland–Altman plot (difference plot) is a method of data plotting that covers better the agreement between two different methods. It can also be used to compare a new measurement technique or method with a gold standard, as even a gold standard does not—and should not—imply it to be without error.
Response 2: We performed a Bland-Altman test to compare imaging methods and we elaborated it in the revised manuscript.
Point 3: In this respect, also the inter-rater agreement is difficult to follow. It is unclear what measurement of estimate the investigators used for their analysis. By only giving test results it is impossible to appreciate what the investigators have done or whether it makes sense.
Response 3: We revised this aspect in the manuscript as well.

Reviewer 2 Report
At the outset, I would like to congratulate the authors for conducting this study. This retrospective study depicts a comparison of US and CT findings in cases of WT diagnosed and managed over a 10-year study period.
In a very limited cohort of 23 patients, Ultrasonography was found to be superior to the CT examination when approximating the antero-posterior and transverse diameters, the computed tomographic examination is cited as superior for estimating the invasion of nephroblastomas. In their clinical setting, the authors also show that Ultrasound is effective in detecting the rupture of the renal capsule, tumor calcifications and invasion of the renal vein, pelvis and ureter.
I have many concerns regarding this paper:
Introduction: The section lacks a study hypothesis. What did you hypothesize before conducting this research?
The introduction section ends abruptly. Please introduce the background of our research work properly.
Methods: What about the approval by the Ethics committee? Did you take it? Was it waived? Please mention.
Results: I am still unsure about the clinical translation of this paper. As per your findings, I understand that US correlates better than CT in terms of predicting the AP and transverse diameters. Staging does not depend on size? Then, what is the clinical translation of this paper?
-You have also stated that CT can never be replaced: as it is better at depicting fat planes, lymphadenopathy, rupture, extension to adjacent organs, etc. Also, these cases get a CT chest always as a part of preoperative workup. The US, on the other hand, is operator-dependent.
-The figures in this paper lack a description. The description should make the figures self-explanatory. Please add it in legends.
-What is depicted in Table 3. There is no caption.
-The statement that "Ultrasound has been shown to be effective in detecting the rupture of the renal capsule, tumor calcifications and invasion of the renal vein, pelvis and ureter" is inaccurate. You have not done a spearman correlation like the AP and transverse diameters. You have just done inter-rater reliability. Kappa value can just say the degree of agreement between the two observers, not the reliability of imaging findings viz-a-viz intraoperative findings. What if the two radiologists are trained at the same center and practice at the same center? Please correct it.
Discussion:
-The discussion of this paper is very general. The authors need to remove the general part about WT, be very specific, and discuss the findings of their research work.
-Please add a paragraph on the limitations of this research work. It is lacking.
Author Response
Reviewer 2
At the outset, I would like to congratulate the authors for conducting this study. This retrospective study depicts a comparison of US and CT findings in cases of WT diagnosed and managed over a 10-year study period. In a very limited cohort of 23 patients, Ultrasonography was found to be superior to the CT examination when approximating the antero-posterior and transverse diameters, the computed tomographic examination is cited as superior for estimating the invasion of nephroblastomas. In their clinical setting, the authors also show that Ultrasound is effective in detecting the rupture of the renal capsule, tumor calcifications and invasion of the renal vein, pelvis and ureter. I have many concerns regarding this paper:
Point 1: Introduction: The section lacks a study hypothesis. What did you hypothesize before conducting this research?
Response 1: In this study we aim to elucidate the advantages of ultrasonography compared to computed-tomography in patients with nephroblastoma.
Point 2: The introduction section ends abruptly. Please introduce the background of our research work properly.
Response 2: Thank you, we will complete this section as you suggest.
Point 3: Methods: What about the approval by the Ethics committee? Did you take it? Was it waived? Please mention.
Response 3: Ethical committee approval was obtained; we will attach it at the end of this paper.
Point 4: Results: I am still unsure about the clinical translation of this paper. As per your findings, I understand that US correlates better than CT in terms of predicting the AP and transverse diameters. Staging does not depend on size? Then, what is the clinical translation of this paper?
Response 4: Nephroblastoma staging does not depend on tumor size, but on the local and distant extension of the tumor, bilaterality and if it was biopsied prior surgery. The importance of the tumor dimensions is expressed in terms of prognosis. The clinical translation of the paper consists in the evaluation of the possibility of using a non-invasive, non-irradiating method as the initial imagistic tool for the diagnosis and staging of nephroblastoma or whether it is of greater sensibility in the evaluation of the essential imagistic characteristics of nephroblastomas, such as tumor dimension, bilaterality, vascular invasion, basinetal or ureteral extension, renal capsule involvement, invasion of adjacent organs, lymph node involvement and the presence of metastases. We will add a paragraph in the manuscript explaining this sentence.
Point 5: You have also stated that CT can never be replaced: as it is better at depicting fat planes, lymphadenopathy, rupture, extension to adjacent organs, etc. Also, these cases get a CT chest always as a part of preoperative workup. The US, on the other hand, is operator-dependent.
Response 5: Yes, we agree that CT cannot be replaced as it is essential for correct and objective stadialization of the tumor.
Point 6: The figures in this paper lack a description. The description should make the figures self-explanatory. Please add it in legends.
Response 6: We will do as you suggest in the revised manuscript. Thank you!
Point 7: What is depicted in Table 3. There is no caption.
Response 7: We will correct this mistake as well.
Point 8: The statement that "Ultrasound has been shown to be effective in detecting the rupture of the renal capsule, tumor calcifications and invasion of the renal vein, pelvis and ureter" is inaccurate. You have not done a spearman correlation like the AP and transverse diameters. You have just done inter-rater reliability. Kappa value can just say the degree of agreement between the two observers, not the reliability of imaging findings viz-a-viz intraoperative findings. What if the two radiologists are trained at the same center and practice at the same center? Please correct it.
Response 8: Thank you for your observation, we will correct it in the revised manuscript.
Point 9: Discussion: The discussion of this paper is very general. The authors need to remove the general part about WT, be very specific, and discuss the findings of their research work.
Response 9: Thank you, we will consider you suggestion.
Point 10: Please add a paragraph on the limitations of this research work. It is lacking.
Response 10: We will add the paragraph in the manuscript.

Round 2
Reviewer 1 Report
The manuscript has improved substantially.
One general remark I would like to make is that the accuracy of many figures is way too precise: 1 patient 4.34% (should be 4) and average age as 45.42 months (should be 45) etc.
Author Response
Thank you for your suggestions and for your help!
Reviewer 2 Report
In the revised manuscript, the authors have answered some of my queries. However, the following need to be added before the manuscript can be reconsidered:
-Please submit the revised manuscript with the corrections highlighted or marked in a different colour. In your point-by-point response you must mention the line number and the paragraph number where the corrections have been made.
-Please add a valid hypothesis at the end of the Introduction section. You need not to answer me but incorporate in your manuscript.
-You have said that you will attach the Ethics committee approval at the end of the paper. Where is that? Please provide the reference number.
-I cannot see the explanation to point no 4. Please highlight it as said it above.
-Figures 9,10,11 still lack a description or a caption. In addition, the legends of other figures need to be revised. Please take the help of a statistical consultant or a senior author.
-How do I know that you made changes to the discussion section in response to point no 9? Please mention the line and paragraphs which have been modified and highlight the.
Author Response
In the revised manuscript, the authors have answered some of my queries. However, the following need to be added before the manuscript can be reconsidered:
Point 1:
Please submit the revised manuscript with the corrections highlighted or marked in a different colour. In your point-by-point response you must mention the line number and the paragraph number where the corrections have been made.
Response 1: We will upload the revised manuscript properly highlighted as you suggested.
Point 2:
Please add a valid hypothesis at the end of the Introduction section. You need not to answer me but incorporate in your manuscript.
Response 2: As you suggest, we will formulate a more accurate hypothesis and introduce it in the introduction section of our manuscript.
Point 3:
You have said that you will attach the Ethics committee approval at the end of the paper. Where is that? Please provide the reference number.
Response 3: The ethical committee approval was on the last page of the point-by-point responses document. There must have been a problem with it. You can find it attached. The registration number is 17335/22.06.2022.
Point 4:
I cannot see the explanation to point no 4. Please highlight it as said it above.
Response 4: You can find the explanation in the last paragraph of the introduction. We highlighted it as you suggested.
Point 5:
Figures 9,10,11 still lack a description or a caption. In addition, the legends of other figures need to be revised. Please take the help of a statistical consultant or a senior author.
Response 5: We will revise the figure captions.
Point 6:
How do I know that you made changes to the discussion section in response to point no 9? Please mention the line and paragraphs which have been modified and highlight them.
Response 6: We highlighted the changes as you suggested. Thank you!
